# RAAG: Ratio Aware Adaptive Guidance

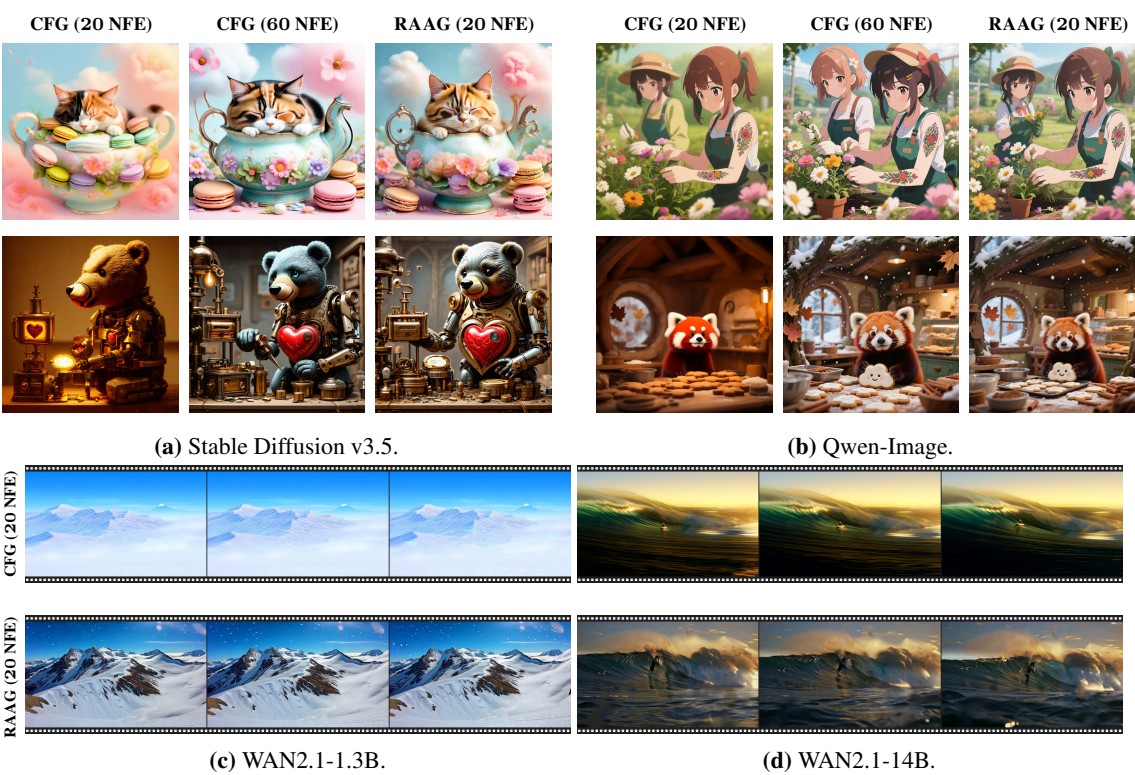

**(a)** Stable Diffusion v3.5.

**(b)** Qwen-Image.

**(c)** WAN2.1-1.3B.

**(d)** WAN2.1-14B.

**Figure 1: Superior efficiency of RAAG:** Our method achieves comparable quality to 60-NFE (30-step) CFG image generation with only 20 NFE (10 steps), demonstrating a $3.0\times$ **speedup** over the standard CFG sampling.

## Abstract

Flow-based generative models have achieved remarkable progress, with classifier-free guidance (CFG) becoming the standard for high-fidelity generation. However, the conventional practice of applying a strong, fixed guidance scale throughout inference is poorly suited for the rapid, few-step sampling required by modern applications. In this work, we uncover the root cause of this conflict: a fundamental sampling instability where the earliest steps are acutely sensitive to guidance. We trace this to a significant spike in the ratio of conditional to unconditional predictions, a spike that we prove to be an inherent property of the training data distribution itself, making it an almost inevitable challenge. Applying a static high guidance value during this volatile initial phase leads to an exponential amplification of the error, degrading image quality. To resolve this, we propose a simple, theoretically

grounded, adaptive guidance schedule that automatically dampens the guidance scale at early steps based on the evolving RATIO. Our method is lightweight, incurs no inference overhead, and is compatible with standard frameworks. Experiments across state-of-the-art image (SD3.5, Qwen-Image) and video (WAN2.1) models show that our approach enables up to 3x faster sampling while maintaining or improving quality, robustness, and semantic alignment. Our findings highlight that adapting guidance to the sampling process, rather than fixing it, is critical for unlocking the full potential of fast, flow-based model.

# 1 INTRODUCTION

Over the past few years, flow-based generative models (Liu et al., 2022; Lipman et al., 2022) have advanced rapidly, offering new ways to synthesize high-quality images (Dhariwal and Nichol, 2021; Huang et al., 2024; Liang et al., 2024; Süleyman and Biricik, 2025; Xiao et al., 2024), videos (Team, 2025), and even multimodal content (Reuss et al., 2024; Ma et al., 2024; Müller-Franzes et al., 2023). Because the sampling trajectory is governed by a deterministic ODE rather than a sequence of stochastic denoising steps, it can be integrated with a high-order ODE solver in only tens of function evaluations (often an order of magnitude lower than the hundreds required by diffusion-based models) while simultaneously eliminating stochastic variance and preserving fidelity.

Much of this progress in both speed and controllability can be attributed to the development of classifier-free guidance(CFG) (Dhariwal and Nichol, 2021; Ho and Salimans, 2022). By blending predictions from both unconditional and conditional predictions, CFG enables models to strongly follow user prompts or target labels, unlocking fine-grained control over output style, content, and meaning. As CFG has become standard, many researchers have sought to push it even further. A variety of modified guidance techniques and adaptive schedules have been proposed to better balance sample diversity, faithfulness, and quality. These approaches, such as *Guidance Scheduler (Wang et al., 2024), $\beta$-CFG (Malarz et al., 2025) and CFG-Zero* (Fan et al., 2025b)*, typically focus on how to adjust the guidance strength, add constraints, or exploit extra side-information. Yet, few works have considered a more fundamental question: *how does the application of guidance weighting affect different temporal phases, especially in fast flow-based models?*

We uncover a previously overlooked weakness in sampling loops of flow-based models: the very first sampling steps are strikingly sensitive to the guidance scale. At the initial noise state, the relative magnitude of the conditional velocity to the unconditional one, defined as RATIO $\rho$, is numerically non-negligible (about 0.1-0.3). Because standard pipelines apply a fixed guidance scale $w$ throughout sampling, their effective control term of sampling errors is *product $w\rho$*. A lower-bound analysis shows that the trajectory error can grow at a rate of roughly $\exp(w\rho)$. Therefore, large $w\rho$ quickly amplifies any numerical or stochastic perturbation, producing semantic drift and visual artifacts. In contrast, reducing RATIO or equivalently moderating $w$ when RATIO is high substantially stabilizes the early trajectory. Controlled experiments in which we vary only the initial sampling noise confirm the prediction: Initial noises with lower-RATIO yield markedly higher ImageReward and CLIPScore, while everything else is kept constant.

To mitigate this vulnerability, we introduce a RATIO-*aware* guidance schedule. Rather than keeping $w$ fixed for the entire trajectory, we recompute it at each sampling step using a lightweight exponential map of the current RATIO, which is $w(\rho_t) = 1 + (w_{\max} - 1)\exp(-\alpha\,\rho_t)$. This closed-form rule mirrors the trend suggested by the error-bound analysis and is further confirmed by our greedy $N$-step search: it retains the head-room of a large guidance scale when the conditional signal is weak ($\rho_t \approx 0$), yet rapidly attenuates guidance when the RATIO spikes. The schedule is *parameter-free at test time*, adds negligible overhead, and can be plugged into any Rectified Flow pipeline without retraining.

Our adaptive guidance schedule enables substantial acceleration of conditional generation without sacrificing sample quality, as measured by ImageReward, CLIPScore, and vBench. Specifically, we achieve up to

$3\times$ faster inference on SD3.5 (Esser et al., 2024), $4\times$ on Lumina (Gao et al., 2024), and $2\times$ on WAN2.1-14B (Team, 2025), each time matching or exceeding the performance of much slower baselines. Furthermore, results are robust to hyperparameter settings and generalize across model scales, datasets, and both image and video domains. As an added benefit, our approach even yields modest quality improvements in some regimes (e.g., higher video quality for WAN2.1-1.3B than WAN2.1-14B), further underscoring its practical effectiveness and versatility.

In summary, our main contributions are as follows:

- We provide the first in-depth, theoretically grounded analysis of why early-step guidance is so impactful in flow-based models, supporting our findings with both toy examples and real datasets.

- We introduce a practical, RATIO-aware adaptive guidance schedule that automatically unifies guidance strength across early and late sampling steps.

- Through extensive experiments on image and video generation, including ablations and hyperparameter studies, we show that our approach enables significant speedup and quality gains over current state-of-the-art, with broad compatibility and easy adoption.

## 2 PRELIMINARY

### 2.1 CONDITIONAL GENERATION OPTIMIZATION

A variety of methods have aimed to improve guidance in diffusion models: **CFG++** (Chung et al., 2024) treats guidance as manifold-constrained inverse problems; **CFG Schedulers** (Xi et al.) optimizes time-dependent strength; **Apply Guidance in Interval** (Kynkäänniemi et al., 2024) restricts guidance to specific noise levels; **Adaptive Guidance** (Castillo et al., 2025) dynamically selects conditional/unconditional paths. **ReCFG** (Xia et al., 2025) corrects expectation shifts via closed-form solutions, while **TFG** (Ye et al., 2024) unifies gradient-based parameterizations. **S²-Guidance** (Chen et al., 2025) leverages stochastic block-dropping for modules introducing empirically low-quality predictions, while **REG** (Gao et al., 2025) theoretically identifies scaled joint distribution as an effective objective for better performance of guidance methods. **FBG** (Koulischer et al., 2025) also introduces a novel paradigm for dynamically adapting guidance scales through the feedback of conditional informativeness. Particularly relevant, **CFG-Zero\*** (Fan et al., 2025b) targets flow-matching models (Esser et al., 2024; Lipman et al., 2022; Fan et al., 2025a; Liu et al., 2022; Gao et al., 2024) by compensating early-step velocity errors on SD3.5 and WAN2.1 (Team, 2025). While these methods deliver progressive improvements in controllability and detail preservation, our work diverges by fundamentally analyzing the underlying dynamics of guidance.

### 2.2 CLASSIFIER FREE GUIDANCE IN RECTIFIED FLOW

In the *Rectified Flow* (RF) framework, a linear interpolation is defined between a data point $x_1 \sim p_{\text{data}}$ and a Gaussian latent variable $x_0 \sim \mathcal{N}(0, I)$ as $x_t = tx_1 + (1-t)x_0$, for $t \in [0, 1]$. The model learns a velocity vector that guides this transition, with two predictors: the *unconditional velocity* $v_u(x_t) = \mathbb{E}[x_0 - x_1 \mid x_t, \varnothing]$, which captures the expected direction without conditioning, and the *conditional velocity* $v_c(x_t, c) = \mathbb{E}[x_1 - x_0 \mid x_t, c]$, incorporating supplementary information $c$ (e.g., class labels or text prompts). To balance quality and conditioning, *Classifier-Free Guidance* (CFG) combines them via:

$$v_{\text{cfg}}(x_t, c) = v_u(x_t) + w \cdot \big(v_c(x_t, c) - v_u(x_t)\big),$$

where $w > 1$ controls the guidance strength. A higher $w$ enhances condition alignment at potential diversity cost, while a lower one promotes diversity.

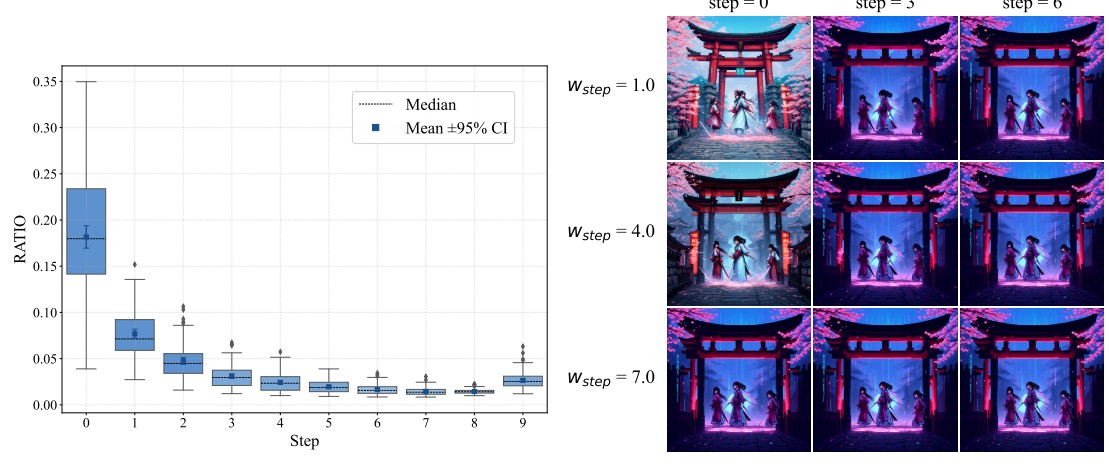

**(a)** RATIO distribution by step.            **(b)** Varying guidance at different steps.

**Figure 2: (a) Dynamics of the conditional–unconditional RATIO $\rho$ during sampling**. The RATIO $\rho$ starts high, which indicates strong conditional influence, but rapidly decays within a few steps and stabilizes. This early spike motivates our RATIO-adaptive guidance schedule. **(b) Effect of varying guidance scales at different steps on image quality in a 10-step constant CFG sampling.** When varying the guidance scale at a single step while keeping it fixed at 7.0 elsewhere, the first step is most sensitive to such changes. Quality degrades once the initial guidance scale $w_0$ exceeds a threshold, revealing over-steering phenomenon from excessive initial guidance.

## 3 RATIO AWARE ADAPTIVE GUIDANCE

Prior to formalizing our method, we introduce two central quantities that underpin our theoretical and empirical study to enable a precise analysis of how conditioning information modulates the sampling dynamics:

**Definition 3.1.** The *velocity gap* between conditional and unconditional velocity predictions is defined as:

$$\delta(x_t, c) := v_c(x_t, c) - v_u(x_t). \tag{1}$$

**Definition 3.2.** The RATIO is defined as:

$$\text{RATIO}(x_t, c) := \frac{\|\delta(x_t, c)\|_2}{\|v_u(x_t)\|_2} = \frac{\|v_c(x_t, c) - v_u(x_t)\|_2}{\|v_u(x_t)\|_2}. \tag{2}$$

Intuitively, Equation (2) measures the relative magnitude of the conditional signal to the unconditional velocity prediction; a larger RATIO indicates stronger prompt influence.

**Section overview.** We first observe a striking sensitivity to the guidance scale in the early reverse steps, characterized by a sharp RATIO spike (Section 3.1). We prove this stems inherently from data distributions (Section 3.2), motivating our adaptive guidance schedule for stability/quality gains (Sections 3.3 and 3.4).

### 3.1 EARLY-STEP GUIDANCE INSTABILITY

We observe that the guidance scales in the early sampling steps disproportionately influence the final output: starting from pure noise, these initial updates determine the semantic trajectory, and excessive guidance at this stage causes persistent semantic drift and artifacts (see Figure 2b). While prior work cautions against

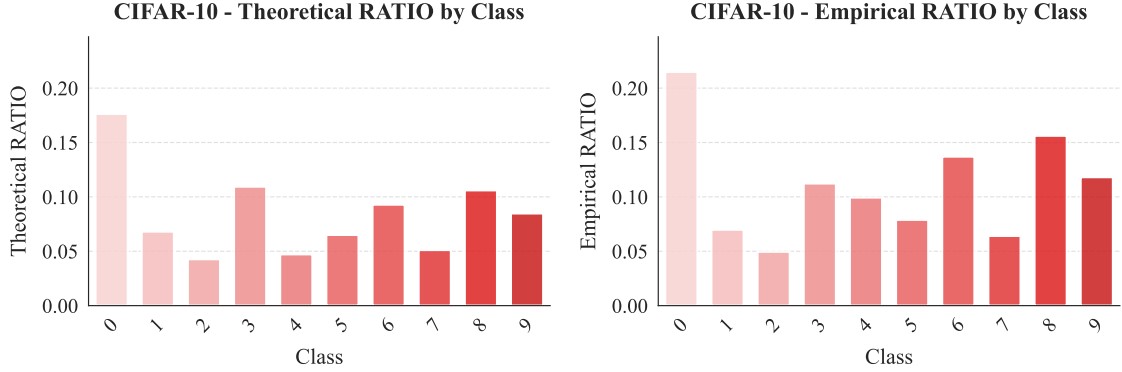

**Figure 3: Agreement between theoretical and empirical first-step RATIO on CIFAR-10.** Quantitative comparison between the closed-form $\rho_{t=0}$ (from $\|\mu_c - \mu_u\|_2/\|x_0 - \mu_u\|_2$) and the empirically measured first-step RATIO across test samples of different classes. The results demonstrate a high degree of consistency between theory and practice, with Pearson correlation coefficient of $r = 0.923$ on CIFAR-10 (see Figure 8 in Appendix B).

high guidance (Sadat et al., 2024), the acute sensitivity of these early steps remains understudied. We trace this instability to a pronounced spike in RATIO between conditional and unconditional velocities during early sampling (see Figure 2a), which amplifies any fixed guidance scale $w$ and makes large values especially disruptive initially. We will later theoretically explain why the initial RATIO is large and how it worsens the instability, highlighting the necessity of a RATIO-aware guidance schedule.

### 3.2  Origin and Inevitability of a Large Initial Ratio

Early-step over-steering and the accompanying ratio spike are two observations that raise a natural question: *why is the ratio so large at the outset?* We answer this by analyzing the RATIO of the initial noise.

**Definition 3.3.** The unconditional data mean is defined as $\mu_u := \mathbb{E}[x_1]$, and the conditional data mean given condition $c$ is defined as $\mu_c := \mathbb{E}[x_1 \mid c]$.

Here, $\mu_u$ denotes the mean of all unconditional data samples, serving as a reference for the overall distribution. In contrast, $\mu_c$ represents the mean of samples matching condition $c$ (e.g., class label or prompt), capturing the shift of the conditional distribution.

**Closed-form expression at $t \to 0$.**  At $t = 0$, the interpolated variable $x_t$ equals the noise sample $x_0$, i.e., $x_t = x_0$. In this case, the unconditional and conditional velocity predictions reduce to expectations:

$$v_u(x_0) = \mathbb{E}\big[x_0 - x_1 \mid x_0\big] = x_0 - \mu_u$$
$$v_c(x_0, c) = \mathbb{E}\big[x_0 - x_1 \mid x_0, c\big] = x_0 - \mu_c \tag{3}$$

Substituting these expressions into the definition of the velocity gap $\delta(x_0, c) = v_c(x_0, c) - v_u(x_0)$, we obtain $\delta(x_0, c) = \mu_u - \mu_c$. Thus, the RATIO at $t = 0$ can be written explicitly as:

$$\text{RATIO}_{t=0}(c) = \frac{\|\delta(x_0, c)\|_2}{\|v_u(x_0)\|_2} = \frac{\|\mu_c - \mu_u\|_2}{\|x_0 - \mu_u\|_2}. \tag{4}$$

Equation (4) indicates that the numerator depends solely on how the condition $c$ shifts the data mean, while the denominator depends on the distance from $x_0$ to the data mean. Since $x_0 \sim \mathcal{N}(0, I)$, the denominator

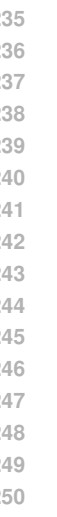
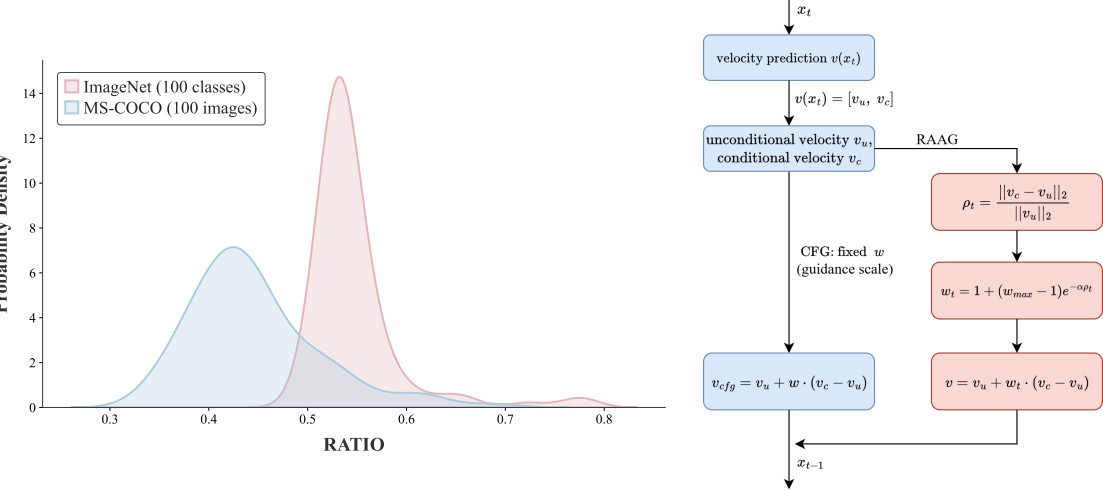

(a) **Distribution of initial RATIO across ImageNet and MS-COCO.**     (b) **Visualization of our method.**

**Figure 4: (a)** Probability density distribution of the first-step conditional-unconditional RATIO $\rho$ for each of the 100 ImageNet100 classes and 100 MS-COCO images. The majority of classes exhibit an initial RATIO concentrated around 0.5, indicating a relatively strong conditional signal at the very first sampling step. **(b)** A compact visualization of RAAG's adaptive guidance scale scheduling based on step-wise RATIOS.

varies little, concentrating near $\sqrt{d}$ (for high dimension $d$, where $d$ denotes the dimension of the noise), with fluctuations diminishing in higher dimensions. The $\mathcal{O}(1)$-scale numerator thus remains significant relative to this stabilized denominator, dominating the initial RATIO.

**Empirical validation.**     We verify the empirical consistency of Equation (4) in Figure 3 and demonstrate that RATIO is a dataset-level property rather than models (Figure 4a). Implementation details are in Appendix C.1.

### 3.3 OVERSIZED RATIO CAUSES SAMPLING COLLAPSE

This subsection presents the exponential amplification of the sampling error introduced by a large initial RATIO, together with an empirical experiment for validation.

**Problem settings.**     Fix a guidance scale $w > 1$ and consider the guided drift

$$v_w(x) \ := \ v_u(x) + w\,\delta(x,c), \qquad \delta(x,c) := v_c(x,c) - v_u(x), \tag{5}$$

where $v_u$ is $L_u$-Lipschitz and $\|\delta(x,c)\|_2 \leq \rho_{\max}\|v_u(x)\|_2$ for all $x \in \mathbb{R}^d$. Let $x(t)$ and $y(t)$ be two solutions of $\dot{x} = v_w(x)$ on the interval $t \in [0,1]$ with initial states $x(0) = x_0$ and $y(0) = y_0$.

**Distance between trajectories.** Define the instantaneous Euclidean separation

$$\Delta(t) \ := \ \big\|x(t) - y(t)\big\|_2, \qquad t \in [0,1]. \tag{6}$$

This quantity measures how initial discrepancies propagate during guided sampling. The lower bound of $\Delta(t)$ can be established in Theorem 3.1 (see Appendix A.2):

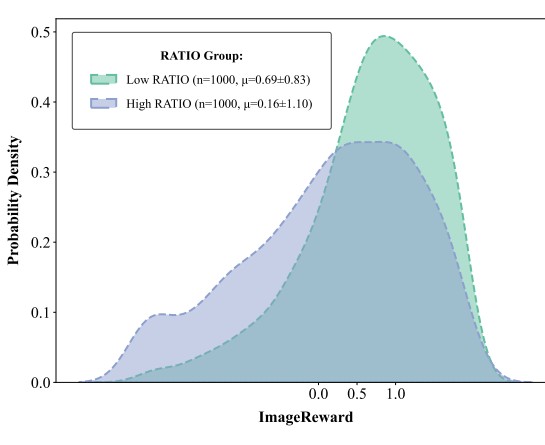 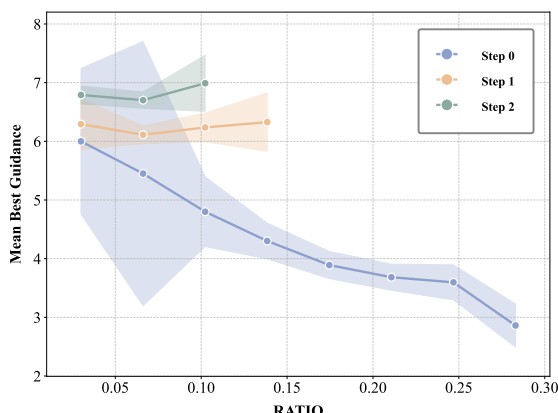

**(a)** Probability density distribution of low and high RATIO groups, respectively.

**(b)** Quantitative relationship between $\rho_k$ and greedy-optimal guidance scales $w_k^\star$ in a 3-step search.

**Figure 5: (a) Probability density of ImageReward scores across different RATIO groups.** KDE plots of ImageReward for low-RATIO versus high-RATIO samples collected at the first sampling step. **(b) Quantitative relationship between** $\rho_k$ **and greedy-optimal guidance scales** $w_k^\star$ **in a 3-step search.** For each step $k < 3$, we plot the measured $\rho_k$ against its corresponding $w_k^\star$ found by greedy tuning. The overall trend closely follows an exponential decay, demonstrating that the optimal guidance schedule across the first few steps can be unified under a single RATIO-based framework.

**Proposition 3.1** (Trajectory Separation Lower Bound). *The Euclidean separation* $\Delta(t)$ *between any two guided sampling trajectories admits the lower bound:*

$$\Delta(t) \; \gtrsim \; \left( \Delta(0) \; - \; \frac{\lambda \|v_w(0)\|_2}{L_w} t \right) \exp\left[ \frac{\lambda \sigma}{(L_u + L_\delta)} \left| \rho - \frac{1}{w} \right| t \right], \tag{7}$$

*where* $L_w$ *and* $L_\delta$ *denote the Lipschitz constants of* $v_w(x)$ *and* $\delta(x,c)$ *respectively, while* $\lambda$ *and* $\sigma$ *represent the positive eigenvalue and minimum singular value of the Jacobian matrix analyzed in Appendix A.1. In addition, the symbol* $\gtrsim$ *denotes "greater than or approximately equal to" here.*

**Role of** $\rho_{\mathbf{max}}$**.** The growth rate of error in Equation (7) is dominated by the exponential factor $\exp\left[ \frac{\lambda \sigma}{L_u + L_\delta} \left| \rho - \frac{1}{w} \right| t \right]$, which could amplify initial perturbations and increase the error lower bound. This bound is minimized when $w = 1/\rho$, suggesting that $w$ should be reduced when the early-step RATIO $\rho$ is large, which is a key insight incorporated into our proposed formulation. When $w\rho_{\max}$ is large, due to high guidance scale $w$ or large RATIO $\rho(x,c)$, the sampling process becomes exponentially sensitive to small perturbations (e.g., floating-point error, stochastic noise, or model mismatch). To validate practical relevance, we group samples by their first-step RATIO and compare ImageReward scores, showing that lower RATIO values reduce error amplification and yield more stable and higher-quality generations.

**Empirical validation.** The results in Figure 5a validate the significant higher performance for low-RATIO noises (0.69±0.83 vs 0.16±1.10, $p < 0.0005$). See Appendix C.2 for implementation details.

### 3.4 RATIO-AWARE ADAPTIVE GUIDANCE

Although we have obtained $w = \frac{1}{\rho}$ in Equation (7) to minimize the lower bound of the sampling error, this theoretically optimal solution does not necessarily lead to better generation in practice, as a lower bound

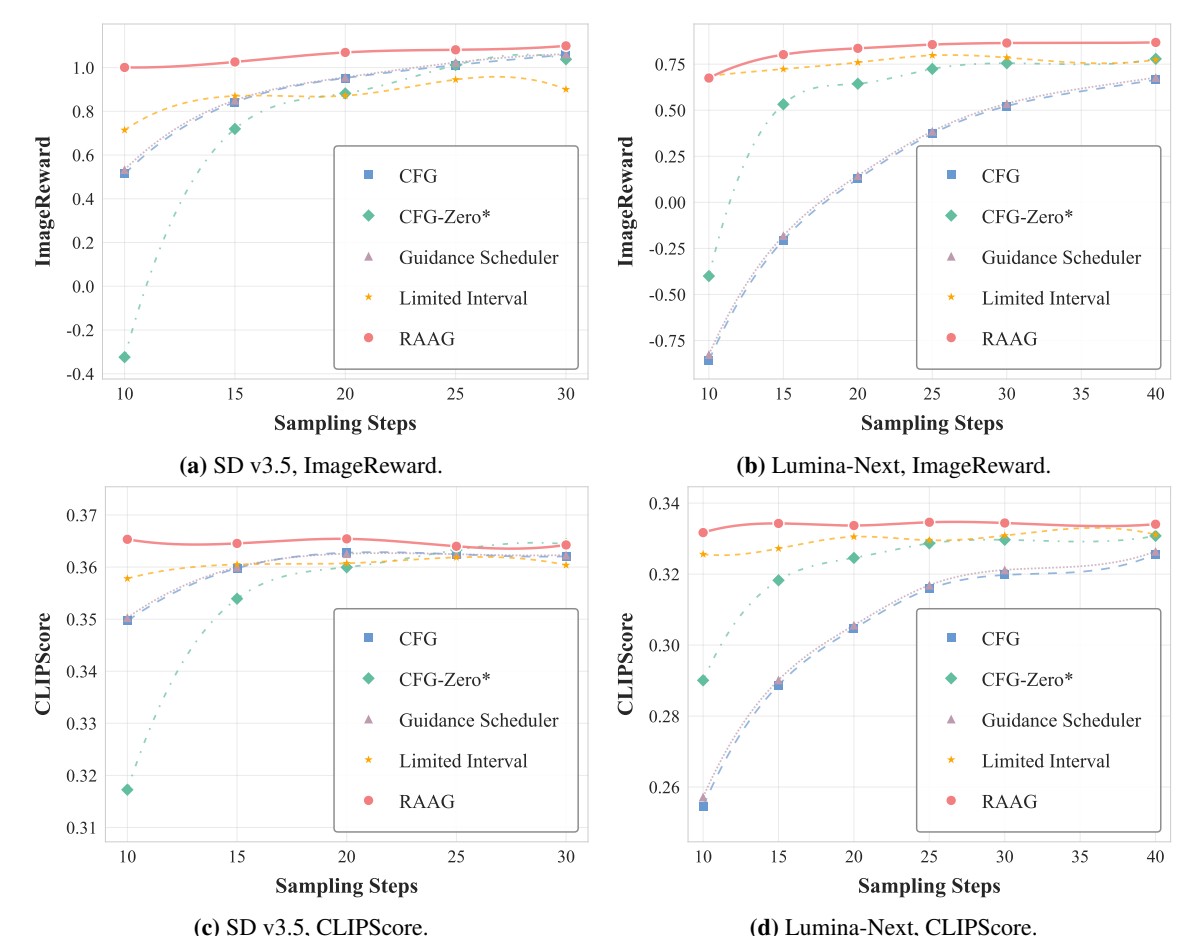

**Figure 6:** Quantitative results of comparative analysis with other baselines in text-to-image generation measured by ImageReward and CLIPScore. The results demonstrate RAAG's remarkable superiority over others in low-step generation, achieving the quality of 30-step generation of CFG in only **10 steps.**

does not guarantee reduced actual sampling error. Moreover, this inverse function form is highly sensitive to velocity prediction. Since $\text{Var}(1/\rho) \propto 1/\rho^4$, the variance increases drastically when $\rho$ is small, which can lead to excessive guidance scale and cause artifacts in generated samples (see Figure 9). To address these issues, we develop a numerically stable formulation that preserves the general trend of $w \propto \frac{1}{\rho}$ while mitigates high-magnitude numerical fluctuations with bounded variance, which is derived from the empirical greedy $N$-step search strategy that determines the greedy optimal guidance scales for the first $N$ sampling steps.

**Greedy $N$-step search.** We tune guidance scales only for the first $N$ steps (later steps by default $w = 7.0$. $N = 3$ in Figure 5b). Algorithm 1 details our greedy search that evaluates $17N$ scale candidates per prompt-step via 10-step sampling. This searching strategy includes the following key properties: (1) Candidate sets always include current values, ensuring non-decreasing quality; (2) Monotonic improvement guarantees stable convergence to local optima. For each step $k < N$, we record $\rho_k = \text{RATIO}(x_{t_k}, c)$ and the optimal scales $w_k^\star$. Aggregating $1,000$ prompts reveals the $\rho_k$-$w_k^\star$ functional trend, as shown in Figure 5b.

**Exponential fit.** Empirically, we observe that $w_k^\star$ exhibits an **exponential decay** with respect to $\rho$ (see Figure 10):

$$w(\rho) = 1 + \big(w_{\max} - 1\big) \exp(-\alpha\rho), \qquad \rho = \text{RATIO}(x_t, c). \qquad (8)$$

where the ceiling $w_{\max}$ safeguards stability by enforcing grid bounds when $\rho \approx 0$, while the decay rate $\alpha$ governs the guidance attenuation speed. This resulting schedule is RATIO-aware, large enough to leverage a weak conditional signal, yet automatically damped when the RATIO spike makes the guidance dangerous.

## 4 EXPERIMENTS

In this section, we conduct comprehensive experiments to evaluate the effectiveness, generalizability, and efficiency of RAAG across both image and video generation domains.

### 4.1 PERFORMANCE ON TEXT-TO-IMAGE GENERATION

We first validate the effectiveness of RAAG in text-to-image tasks. Specifically, to further demonstrate the generality of our approach, we apply RAAG to leading text-to-image diffusion models including Stable Diffusion v3.5 (SD3.5) and Lumina-Next. We assess generation quality using common metrics such as ImageReward and CLIPScore, comparing our method against the corresponding baselines. Experimental results in Figure 6 show that RAAG consistently improves generation quality and semantic alignment, achieving a $3\times$ speedup equivalent to 30-step generations.

### 4.2 PERFORMANCE ON TEXT-TO-VIDEO GENERATION

We apply our RATIO-aware adaptive guidance to recent video diffusion frameworks and measure the quality of generated videos using standard metrics, including Imaging Quality and Aesthetic Quality in vBench. Results in Figure 12 demonstrate that our method significantly enhances both visual coherence and semantic consistency of generated videos compared to baseline methods, even surpassing WAN 14B under the evaluation of Imaging Quality while only using WAN 1.3B.

### 4.3 ABLATION STUDIES AND PARAMETER SENSITIVITY

We conduct comprehensive ablation studies to justify our design choices and parameter selections as follows: **(1) Guidance Modeling:** We compare exponential decay against linear, inverse proportional (see Theorem 3.1), and sigmoid functions. Exponential decay outperforms other formulations empirically. **(2) Hyperparameter Sensitivity:** We analyze the effect of $w_{\max}$ and the decay rate $\alpha$. RAAG shows robust performance across reasonable ranges, confirming practical utility. **(3) Scheduler Generalization:** Applied to UniPC, RAAG maintains consistent gains (see Figure 15), demonstrating broad applicability. Detailed quantitative results of these ablations can be found in Appendix C.6.

## 5 CONCLUSION

We propose RAAG, a training-free guidance schedule derived from the critical RATIO-spike phenomenon. Without retraining or architectural changes, it consistently outperforms existing guidance schedules across flow-based models with minimal overhead, particularly in low-step regimes. Despite significant gains, RAAG faces two core constraints: **(1) Limited to Rectified Flow:** While effective in RF-based pipelines, RAAG does not consistently improve diffusion-based models (see Table 3). Generalizing the approach to both paradigms remains future work. **(2) Limited Gains in High-Step Regimes:** RAAG significantly improves low-step generation, but gains are modest beyond $40$ steps. We attribute this to the inherent model corrections compensate for earlier suboptimal conditions.

**Ethics Statement.** We, the authors, have read and adhere to the ICLR Code of Ethics. Our work introduces a training-free guidance schedule to improve the efficiency and quality of flow-based generative models. The research is technical in nature and does not involve the collection of new datasets or direct interaction with human subjects. All experiments were conducted using publicly available models and datasets, including Stable Diffusion v3.5, Lumina-Next, Qwen-Image, WAN2.1, ImageNet, MS-COCO, TempoFunk and CIFAR-10. While this paper's primary contribution is a technical advancement in the field of generative models, we recognize that the broader application of such models can raise ethical concerns, including the generation of harmful, biased, or misleading content. This work is not intended to address these broader issues, but rather to improve the efficiency of the underlying technology. We hope that our contribution will be used to advance research in a responsible and ethical manner.

**Reproducibility Statement.** All theories proposed and all experimental results reported in this work are reproducible. The detailed derivation and the underlying assumptions for Theorem 3.1 are presented in Appendix A, which provides a clear, step-by-step theoretical analysis. For the experiments described in Section 4, the core code implementing our method is available in Appendix E.

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

# A DERIVATION

## A.1 APPROXIMATE LOWER BOUND OF THE TRAJECTORY DIFFERENCE

In this section, we provide a detailed derivation of the lower bound shown in Equation (7).

For the solutions $x(t)$ and $y(t)$ of $\dot{x} = v_w(x)$, let $z(t) = x(t) - y(t)$. Assume $\Delta(t_0)$ is sufficiently small and positive. Shift time so that $t = t + t_0$ and $t_0 = 0$, hence $t \to 0$ and $z(t) \to z(0)$. Near $t = 0$ the dynamics are linearized by

$$\dot{z}(t) \approx J_{v_w} z(t), \tag{9}$$

where $J_{v_w} = J_{v_w}(y)$ is the Jacobian of $v_w$ at $y$. With $\Delta(t) = \|z(t)\|_2$ and $u(t) = z(t)/\Delta(t)$,

$$\dot{\Delta}(t) \approx \Delta(t)\, u(t)^\top J_{v_w}\, u(t). \tag{10}$$

**Jordan solution.** Let $J$ be the Jordan form of $J_{v_w}$ with eigenvalues $\lambda_1, \ldots, \lambda_m$ (blocks $s_{ij}$, multiplicities $r_i$). The solution of Equation (9) is

$$z(t) = \sum_{i=1}^{m} e^{\lambda_i t} \sum_{j=1}^{r_i} \sum_{k=1}^{s_{ij}} t^{k-1} c_{ijk} e_{ijk}. \tag{11}$$

In reality, the disturbance is random and possibly occur in any direction. To analyze stability, the focus is primarily on the worst-case scenario. Hence, for the sake of analysis, we suppose $z(0)$ lies in the direction of an eigenvector of the matrix $J_{v_w}$ corresponding to a positive eigenvalue $\lambda$, i.e.,

$$J_{v_w} z(t) = \lambda z(t),$$

where $\lambda > 0$. Assume that $\delta(\cdot)$ is $L_\delta$-Lipschitz and $v_w(\cdot)$ is $L_w$-Lipschitz where $L_w = L_u + wL_\delta$ and $L_\delta, L_w > 0$. Then

$$
\begin{aligned}
\dot{\Delta}(t) &\gtrsim \Delta(t) u(t)^\top \lambda u(t) \\
&= \lambda \Delta(t) \\
&= \lambda \|z(t) - 0\|_2 \\
&\geq \frac{\lambda}{L_w} \|v_w(z(t)) - v_w(0)\|_2 \\
&\geq \frac{\lambda}{L_w} \big(\|v_w(z(t))\|_2 - \|v_w(0)\|_2\big) \\
&= \frac{\lambda}{L_u + wL_\delta} \big(\|v_u(z(t)) + w\delta(z(t))\|_2 - \|v_w(0)\|_2\big) \\
&= \frac{\lambda}{L_u + wL_\delta} \Big(\|v_u(z)\|_2 \Big\| \frac{v_u(z)}{\|v_u(z)\|_2} + w\rho \frac{\delta(z)}{\|\delta(z)\|_2} \Big\|_2 - \|v_w(0)\|_2\Big),
\end{aligned} \tag{12}
$$

/

where $\rho = \frac{\|\delta(z)\|_2}{\|v_u(z)\|_2}$ serves as an approximation to $\rho(x) = \frac{\|\delta(x,c)\|_2}{\|v_u(x)\|_2}$. More precisely, $\rho$ can be expressed as:

$$\rho = \frac{\|(v_c(x) - v_u(x)) - (v_c(y) - v_u(y))\|_2}{\|v_u(x) - v_u(y)\|_2}. \tag{13}$$

The approximation in Equation (13) has been validated through experimental data on the properties of initial noises within flow-based models, as shown in Figure 7.

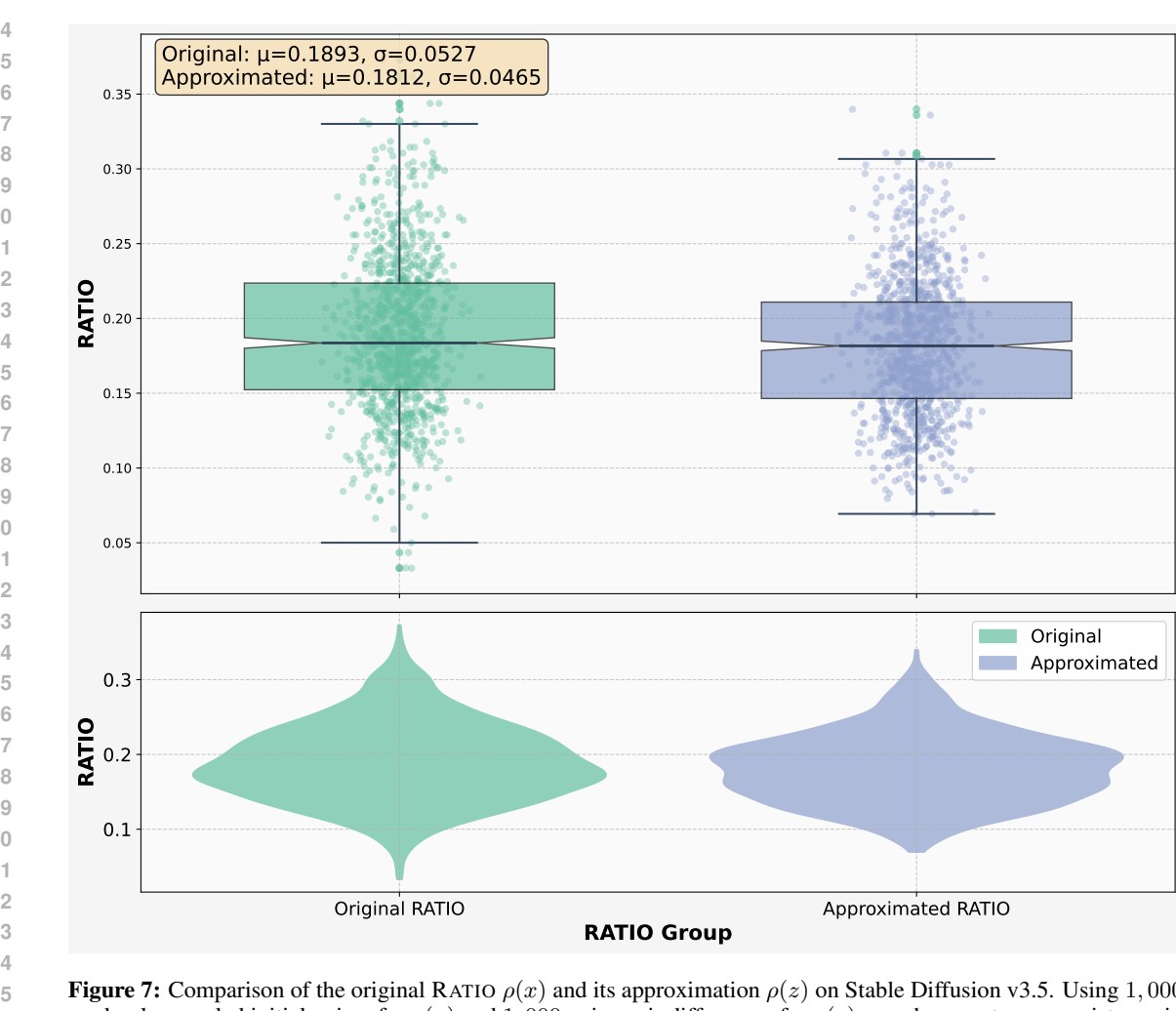

**Figure 7:** Comparison of the original RATIO $\rho(x)$ and its approximation $\rho(z)$ on Stable Diffusion v3.5. Using $1,000$ randomly sampled initial noises for $\rho(x)$ and $1,000$ noise-pair differences for $\rho(z)$, we observe strong consistency in both value and distribution, validating our approximation.

Let $v := \frac{v_u(z)}{\|v_u(z)\|_2}$ and $\delta := \frac{\delta(z)}{\|\delta(z)\|_2}$. Then

$$
\begin{aligned}
\left\| \frac{v_u(z)}{\|v_u(z)\|_2} + w\rho \frac{\delta(z)}{\|\delta(z)\|_2} \right\|_2 &= \|v + w\rho\delta\|_2 \\
&= \sqrt{(v + w\rho\delta)^\top (v + w\rho\delta)} \\
&= \sqrt{\|v\|_2^2 + w^2\rho^2 \|\delta\|_2^2 + 2w\rho\delta^\top v} \\
&\geq \sqrt{(1 - w\rho)^2} \\
&= |1 - w\rho|.
\end{aligned}
\tag{14}
$$

Due to $w \geq 1$, we have

$$L_u + wL_\delta \leq w(L_u + L_\delta).$$

Then

$$\dot{\Delta}(t) \gtrsim \lambda \|v_u(z)\|_2 \frac{|1 - w\rho|}{w(L_u + L_\delta)} - \lambda \frac{\|v_w(0)\|_2}{L_w}$$

$$\gtrsim \lambda\sigma \frac{|1 - w\rho|}{w(L_u + L_\delta)} \Delta(t) - \lambda \frac{\|v_w(0)\|_2}{L_w}, \tag{15}$$

where $J_{v_u} := J_{v_u}(y)$ is the Jacobian matrix of $v_u$ and $\sigma$ is the minimum singular value of $J_{v_u}$ satisfying $\|v_u(z)\| \geq \sigma\Delta(t)$. Note that here we do not consider the trivial case $z(t)$ is in the kernel space of $J_{v_u}$.

Giving

$$\dot{\Delta}(t) \gtrsim A\Delta(t) - B, \qquad A := \frac{\lambda\sigma}{w(L_u + L_\delta)}|1 - w\rho|, \; B := \frac{\lambda}{L_w}\|v_w(0)\|_2. \tag{16}$$

**Integral form (Newton-Leibniz).** By applying the Newton-Leibniz formula to Equation (16), we derive the integral form:

$$\Delta(t) - \Delta(0) \gtrsim \int_0^t A\Delta(s)\,ds - Bt. \tag{17}$$

**Integration (Grönwall).** Applying the Grönwall inequality to Equation (17) yields:

$$\Delta(t) \gtrsim \Delta(0) - Bt + \int_0^t (\Delta(0) - Bs)Ae^{A(t-s)}\,ds. \tag{18}$$

**Bounding the right-hand side.** Given $B \geq 0$ and $0 \leq s \leq t$, we relax the right-hand side of Equation (18) as follows:

$$\Delta(t) \gtrsim \Delta(0) - Bt + \int_0^t (\Delta(0) - Bs)Ae^{A(t-s)}\,ds$$

$$\geq \Delta(0) - Bt + \int_0^t (\Delta(0) - Bt)Ae^{A(t-s)}\,ds \tag{19}$$

$$= (\Delta(0) - Bt)(1 + \int_0^t Ae^{A(t-s)}\,ds).$$

**Differential identity and evaluation.** Using the identity $d(-e^{A(t-s)}) = Ae^{A(t-s)}\,ds$, we simplify:

$$\boxed{\begin{aligned} \Delta(t) &\gtrsim (\Delta(0) - Bt)(1 + \int_0^t Ae^{A(t-s)}\,ds) \\ &= (\Delta(0) - Bt)(1 + \int_0^t d[-e^{A(t-s)}]) \\ &= (\Delta(0) - Bt)(1 + [-e^{A(t-s)}]\Big|_0^t) \\ &= (\Delta(0) - Bt)e^{At}, \end{aligned}} \tag{20}$$

where $A$, $B$ are defined in Equation (16).

**Sensitivity condition.** As $t \to 0$, the first-order approximation yields $\Delta(0) - Bt \approx \Delta(0) \geq 0$, as established in Equation (20). Therefore, we only need to consider the minimum of the amplification coefficient $A = \dfrac{\lambda\sigma}{w(L_u + L_\delta)} |1 - w\rho|$, which is minimized when $|1 - w\rho| \approx 0$, i.e.

$$w \approx \frac{1}{\rho} \quad \left(\rho = \frac{\|\delta(z)\|_2}{\|v_u(z)\|_2}\right),$$

which suppresses exponential growth and limits the sensitivity of $\Delta(t)$ to perturbations.

**Limitations for diffusion-based models.** The empirical RATIO values observed in diffusion-based models (typically exceeding $0.5$) are significantly larger than those in flow-based models. As indicated by the approximation in Equation (13), when $\rho(x)$ and $\rho(y)$ attains high values, the perturbation introduced by $\rho(y)$ becomes non-negligible. This results in a substantial discrepancy between $\rho(x)$ and $\rho(z)$, undermining the underlying assumption of our exponential decay formulation. Consequently, the proposed exponential decay mechanism fails to effectively address the initial RATIO-spike phenomenon in diffusion-based models.

## A.2 UPPER BOUND OF THE TRAJECTORY DIFFERENCE.

In addition to the lower bound, we also derive an upper bound for the trajectory difference for completeness.

**Time derivative of $\Delta(t)$.** Let $z(t) := x(t) - y(t)$ and define the separation $\Delta(t) := \|z(t)\|_2$. Using the chain rule for the Euclidean norm,

$$\frac{d}{dt}\Delta(t) = \frac{z(t)^\top}{\|z(t)\|_2} \frac{d}{dt}z(t) = \frac{z(t)}{\Delta(t)} \cdot \left[v_w\big(x(t)\big) - v_w\big(y(t)\big)\right]. \tag{21}$$

Introduce the unit vector $u(t) := z(t)/\Delta(t)$, so that $\|u(t)\|_2 = 1$. Equation (21) becomes

$$\frac{d}{dt}\Delta(t) = u(t)^\top \left[v_w\big(x(t)\big) - v_w\big(y(t)\big)\right]. \tag{22}$$

For Equation (22), applying the Cauchy–Schwarz inequality $|a^\top b| \leq \|a\|_2 \|b\|_2$ with $a = u(t)$ and $b = v_w(x(t)) - v_w(y(t))$ gives

$$\left|\frac{d}{dt}\Delta(t)\right| \leq \|v_w\big(x(t)\big) - v_w\big(y(t)\big)\|_2. \tag{23}$$

Because $\Delta(t) \geq 0$, the left-hand side of Equation (23) is non-negative, so the absolute value can be dropped to yield the upper bound used later:

$$\frac{d}{dt}\Delta(t) \leq \big\|v_w\big(x(t)\big) - v_w\big(y(t)\big)\big\|_2. \tag{24}$$

**Bounding the velocity difference.** For arbitrary $x, y \in \mathbb{R}^d$,

$$
\begin{aligned}
\big\|v_w(x) - v_w(y)\big\|_2 &\leq \|v_u(x) - v_u(y)\|_2 + w\|\delta(x,c) - \delta(y,c)\|_2 \\
&\leq L_u\|x - y\|_2 + w\big[\|\delta(x,c)\|_2 + \|\delta(y,c)\|_2\big] \\
&\leq L_u\|x - y\|_2 + w\rho_{\max}\big[\|v_u(x)\|_2 + \|v_u(y)\|_2\big] \\
&\leq L_u\|x - y\|_2 + w\rho_{\max}\big[\|v_u(y)\|_2 + L_u\|x - y\|_2 + \|v_u(y)\|_2\big] \\
&= L_u\big(1 + w\rho_{\max}\big)\|x - y\|_2 + 2w\rho_{\max}\|v_u(y)\|_2.
\end{aligned}
\tag{25}
$$

The penultimate line uses the triangle inequality and the Lipschitz property $\|v_u(x)\|_2 \leq \|v_u(y)\|_2 + L_u\|x - y\|_2$.

**Applying the bound to the trajectories.** Substituting $x = x(t)$ and $y = y(t)$ in Equation (25) and combining with Equation (24) yields

$$\frac{d}{dt}\Delta(t) \;\leq\; L_u\big(1 + w\rho_{\max}\big)\,\Delta(t) \;+\; 2w\rho_{\max}\big\|v_u\big(y(t)\big)\big\|_2. \tag{26}$$

**Grönwall preparation.** Because $t \in [0,1]$ is compact and $v_u$ is continuous, by the Weierstrass Extreme Value Theorem the quantity

$$V_{\max} := \sup_{s \in [0,1]} \|v_u\big(y(s)\big)\|_2 \tag{27}$$

is finite. Replacing the last factor in Equation (26) with $V_{\max}$ gives the linear Grönwall form

$$\frac{d}{dt}\Delta(t) \;\leq\; L_u\big(1 + w\rho_{\max}\big)\,\Delta(t) \;+\; 2w\rho_{\max}\,V_{\max}. \tag{28}$$

**Consequences for generative quality.** Solve the linear differential inequality Equation (28) by Grönwall's lemma:

$$\Delta(t) \;\leq\; \Big(2w\rho_{\max}V_{\max}t + \Delta(0)\Big)e^{L_u(1+w\rho_{\max})t}. \tag{29}$$

The upper bound is intrinsically linked to the exponential term with regard to $w\rho$, further demonstrating the inherent instability of the sampling trajectory.

# B  SUPPLEMENTARY FIGURES

## B.1  CORRELATION ON CIFAR-10

We supplement the correlation between theoretical and empirical RATIO as shown in Figure 8.

## B.2  EXAMPLE OF ARTIFACTS

We demonstrate qualitative outcomes of the guidance scheduler $w = \frac{1}{\rho}$ proposed in Theorem 3.1, revealing conspicuous artifacts induced by unregulated excessive scaling.

## B.3  GREEDY N-STEP SEARCH ALGORITHM

We detail the implementation of the Greedy N-Step Search algorithm introduced in Section 3.4, as shown in Algorithm 1.

## B.4  EXPONENTIAL FIT

We supplement the fitting results of exponential decay to the RATIO-guidance data obtained by greedy $N$-step search mentioned in Section 3.4, with results presented Figure 10.

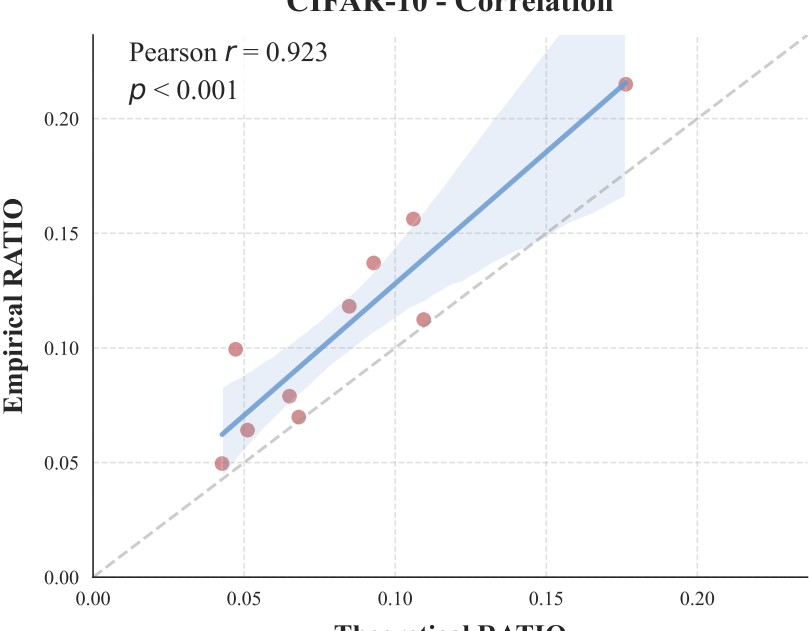

**Figure 8:** The correlation between the closed-form $\rho_{t=0}$ (from $\|\mu_c - \mu_u\|_2/\|x_0 - \mu_u\|_2$) and the empirically measured first-step RATIO as discussed in Section 3.2.

---

**Algorithm 1** Greedy $N$-Step Search

---

**Require:** Default guidance scale $w = 7$, step count $T$, search depth $N$
**Ensure:** Optimized guidance scales $\{w_t^*\}_{t=0}^{N-1}$
 1: **Initialization:**
 2:    Set $w_t \leftarrow 7$ for all steps $t \in \{0, \ldots, T-1\}$
 3:    Define search grid $\mathcal{W} \leftarrow \{1, 1.5, 2, \ldots, 9\}$                             ▷ 0.5 spacing
 4: **for** $t = 0$ **to** $N - 1$ **do**
 5:      **Step-$t$ Search:**
 6:        Fix $\{w_0^*, \ldots, w_{t-1}^*\}$ from previous steps
 7:        $w_t^* \leftarrow \underset{w \in \mathcal{W}}{\arg\max} \, \text{ImageReward}(w_0^*, \ldots, w_{t-1}^*, w, 7, \ldots, 7)$
 8: **end for**
 9: **Return** $(w_0^*, \ldots, w_{N-1}^*)$                                    ▷ Steps $N$ to $T-1$ keep $w = 7$

---

## C  EXPERIMENTS

### C.1  VALIDATION EXPERIMENT ON IMAGENET AND MS-COCO

On ImageNet and MS-COCO, we observe consistently high initial RATIO values after SD3.5 VAE encoding (Figure 4a), confirming this as a dataset-level property rather than model-dependent. In addition, we further trained RF-Transformer baselines on CIFAR-10 to test whether Equation (4) holds in practice. The measured

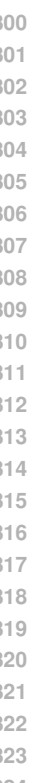

**Figure 9:** Qualitative evaluations of the guidance schedule $w = \frac{1}{\rho}$. The top row displays the original reference images, while the bottom row presents outputs generated under the $w = \frac{1}{\rho}$ scheduling framework. The results reveal conspicuous artifacts attributable to excessive scaling magnitudes.

$\text{RATIO}_{t=0}$ closely matches the pre-computed ideal RATIO (see Figure 3), with a strong linear correlation ($r = 0.923$).

### C.2  VALIDATION EXPERIMENT FOR HIGH/LOW INITIAL RATIO

We conducted controlled single-phase experiments on SD3.5 using 100 MS-COCO prompts (20 initial noises per prompt, $w = 7.0$, 10 steps). Measuring $\rho(x_t, c)$ at the first sampling step, we grouped the initial noises by RATIO: the top/bottom 10 formed high/low-RATIO groups. Results in Figure 5a showed significantly higher performance for low-RATIO groups ($0.69\pm0.83$ vs $0.16\pm1.10$, $p < 0.0005$).

### C.3  TEXT-TO-IMAGE GENERATION

#### C.3.1  MEASUREMENT BY GENEVAL

We further validate RAAG 's effectiveness using GenEval across multiple metrics. As shown in Table 1, RAAG consistently outperforms CFG in most scenarios.

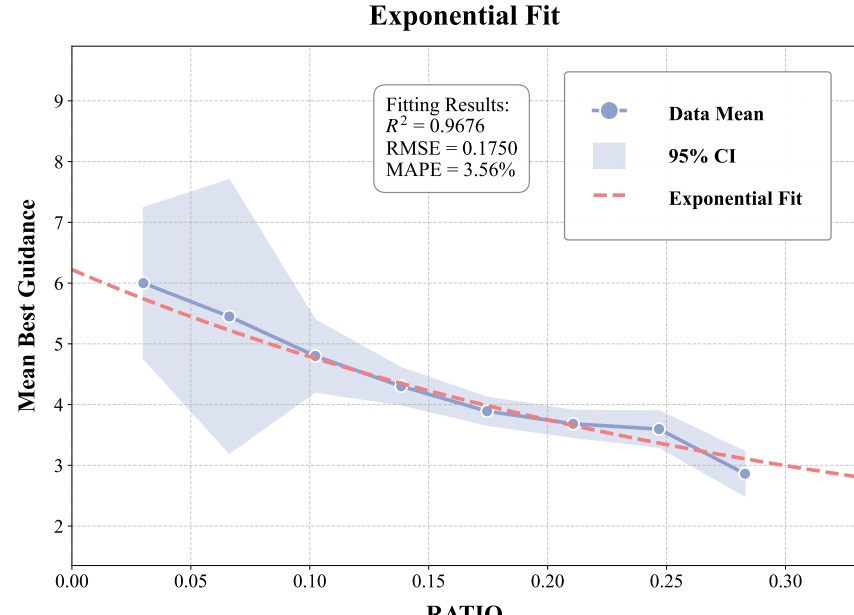

**Figure 10:** Quantitative results of the exponential fit between RATIO and empirically optimal guidance $w$ demonstrate high accuracy, validated by the coefficient of determination ($R^2$), root mean square error (RMSE), and mean absolute percentage error (MAPE).

**Table 1:** Quantitative results of text-to-image generation measured by GenEval. The evaluation demonstrates RAAG's consistent superiority over standard CFG across different dimensions of GenEval, attaining a maximum $+\mathbf{21.25\%}$ **absolute improvement** on the **Single Object** metric.

| Model | Method | Single Object | Two Object | Overall Score |
|---|---|---|---|---|
| SD35 | CFG | 96.25% | 85.00% | 0.9063 |
| SD35 | RAAG | **98.75%** | 85.00% | **0.9188** |
| Lumina-Next | CFG | 71.25% | 35.00% | 0.5313 |
| Lumina-Next | RAAG | **92.50%** | **45.00%** | **0.6875** |

### C.3.2   PERFORMANCE ON QWEN-IMAGE

Qwen-Image (Wu et al., 2025), a recent image generation foundation model within the Qwen series, demonstrates notable improvements in semantic alignment and visual fidelity. While Qwen-Image is already a powerful model, our proposed method, RAAG, further enhances its capabilities. As our experimental data in Figure 11 show, RAAG significantly improves both generation quality and reward scores, pushing the boundaries of what is possible even with state-of-the-art foundation models.

### C.4   TEXT-TO-VIDEO GENERATION

We present the quantitative results of the text-to-video generation in Figure 12.

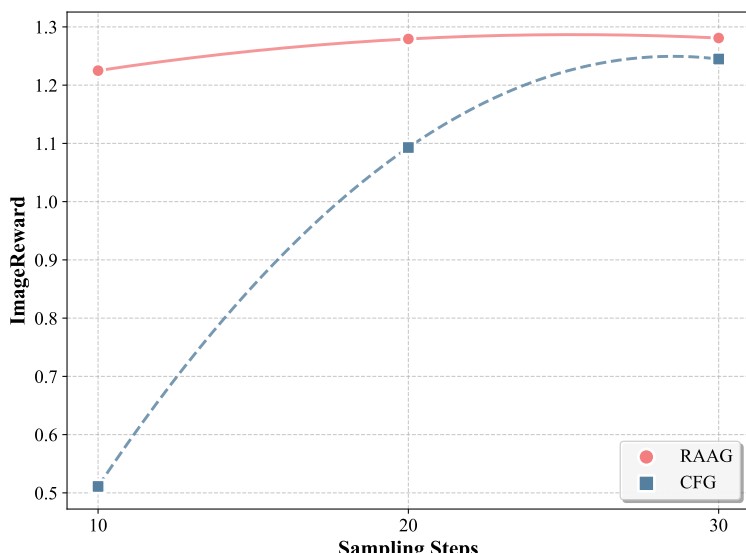

**Figure 11:** Quantitative evaluation of text-to-image generation performance on Qwen-Image. RAAG demonstrates superior performance compared to traditional CFG scheduling, with particularly notable improvements in low sampling-step regimes.

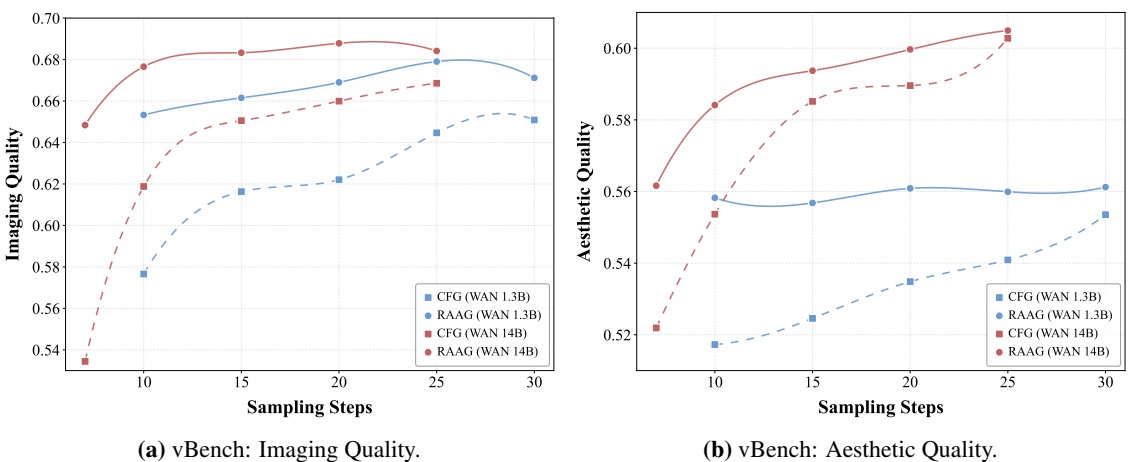

(a) vBench: Imaging Quality.    (b) vBench: Aesthetic Quality.

**Figure 12:** Quantitative results of text-to-video generation, comparing **CFG and RAAG** under vBench. The results strongly validate the effectiveness of RAAG on video generations, even overtaking CFG-WAN 14B just on WAN 1.3B.

## C.5 PERFORMANCE ON SCORE-BASED MODELS

In this section, we assess the applicability of RAAG to Stable Diffusion v2, examining its integration with score-based diffusion models. Preliminary results in Table 3 reveal subpar performance, highlighting limitations in our current approach. This discrepancy merits deeper investigation and will guide our future work.

**Table 2:** Quantitative results of text-to image generation on Stable Diffusion v2, comparing **RAAG** and **CFG**. *Note:* decay rate $\alpha = 12$, guidance ceiling $w_{\max} = 18$.

| Step | ImageReward | | CLIPScore | |
|------|------|------|------|------|
| | RAAG | CFG | RAAG | CFG |
| 10 | $-0.4306$ | $-0.0983$ | 0.3498 | 0.3578 |
| 20 | $-0.2368$ | 0.0814 | 0.3575 | 0.3622 |
| 30 | $-0.1807$ | 0.1463 | 0.3586 | 0.3635 |
| 40 | $-0.1444$ | 0.1702 | 0.3575 | 0.3641 |

**Table 3:** Quantitative results of text-to image generation on Stable Diffusion v2, comparing **RAAG** and **CFG**. *Note:* decay rate $\alpha = 12$, guidance ceiling $w_{\max} = 18$.

| Step | ImageReward | | CLIPScore | |
|------|------|------|------|------|
| | RAAG | CFG | RAAG | CFG |
| 10 | $-0.4306$ | $-0.0983$ | 0.3498 | 0.3578 |
| 20 | $-0.2368$ | 0.0814 | 0.3575 | 0.3622 |
| 30 | $-0.1807$ | 0.1463 | 0.3586 | 0.3635 |
| 40 | $-0.1444$ | 0.1702 | 0.3575 | 0.3641 |

## C.6 ABLATION STUDIES

In this section, we provide supplementary quantitative results from the ablation studies discussed in Section 4.3.

1. **Modeling Function Type (Figure 13):** We have conducted a comparative analysis of modeling functions relating the guidance scale to RATIO, demonstrating the superior performance of our selected formulation.

2. **Hyperparameter Sensitivity (Figure 14):** Through a rigorous grid search over the parameter space, we verify RAAG's insensitivity to hyperparameter selection, with performance variations remaining minor across all tested configurations.

3. **Scheduler Generalization (Figure 15):** We have conducted a further extension of RAAG to UniPC to demonstrate consistent performance across different sampling schedulers, confirming architecture-agnostic applicability.

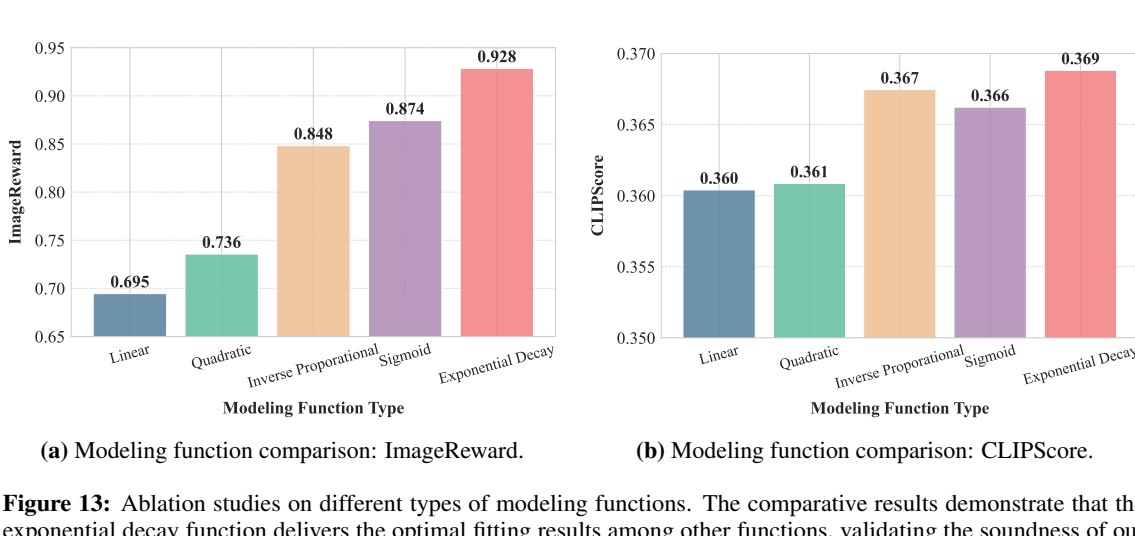

(a) Modeling function comparison: ImageReward.

(b) Modeling function comparison: CLIPScore.

**Figure 13:** Ablation studies on different types of modeling functions. The comparative results demonstrate that the exponential decay function delivers the optimal fitting results among other functions, validating the soundness of our method.

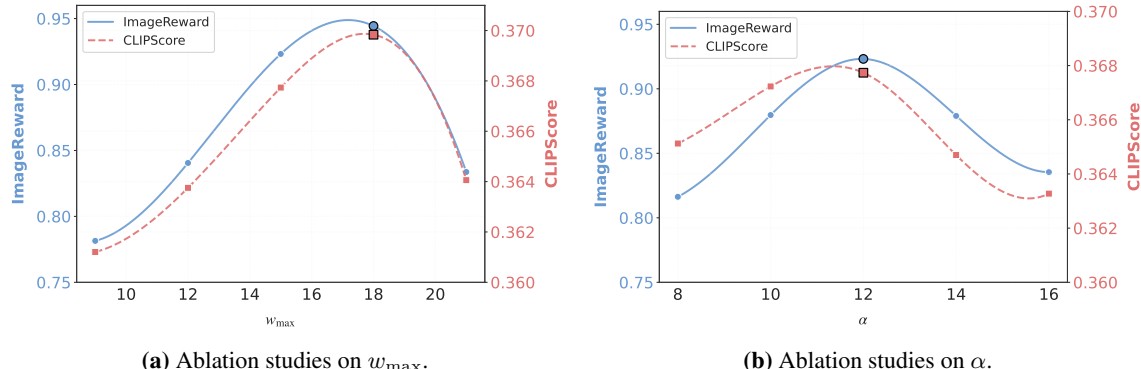

(a) Ablation studies on $w_{\max}$.

(b) Ablation studies on $\alpha$.

**Figure 14:** Ablation studies on hyperparameters $w_{\max}$ and $\alpha$. Figures 14a and 14b present the variation trends of ImageReward and CLIPScore with respect to hyperparameters and validate our optimal selection of $w_{\max}$ and $\alpha$, indicating that the generation quality of RAAG is largely insensitive to hyperparameter selection, with performance sustained across an extensive parameter space.

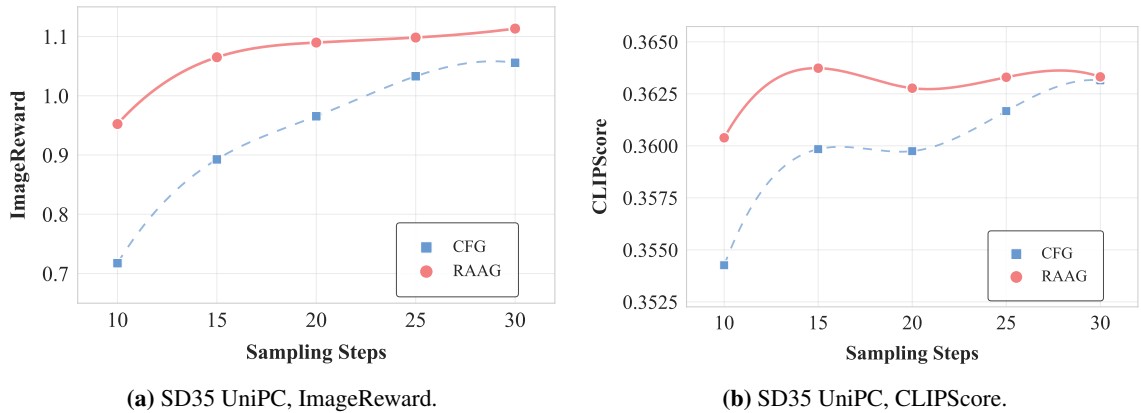

(a) SD35 UniPC, ImageReward.

(b) SD35 UniPC, CLIPScore.

**Figure 15:** Ablation studies of schedulers on UniPC. The comparative results present the consistent superiority of RAAG over CFG across different schedulers, further validating the wide-ranging adaptability of our method.

# D VISUALIZATION RESULTS

In this part, we present additional qualitative results to visually illustrate the performance improvement achieved by RAAG, as shown in Figure 16 and Figure 17.

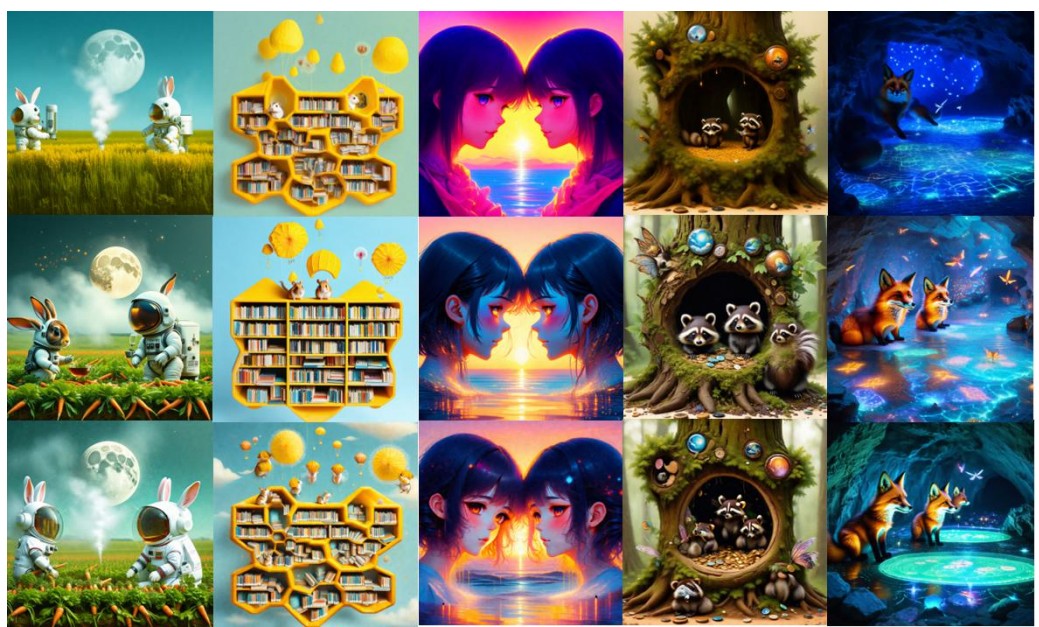

**Figure 16:** Qualitative comparison of text-to-image generation using Stable Diffusion v3.5. From top to bottom: **10-step CFG**, **30-step CFG**, and **10-step RAAG**.

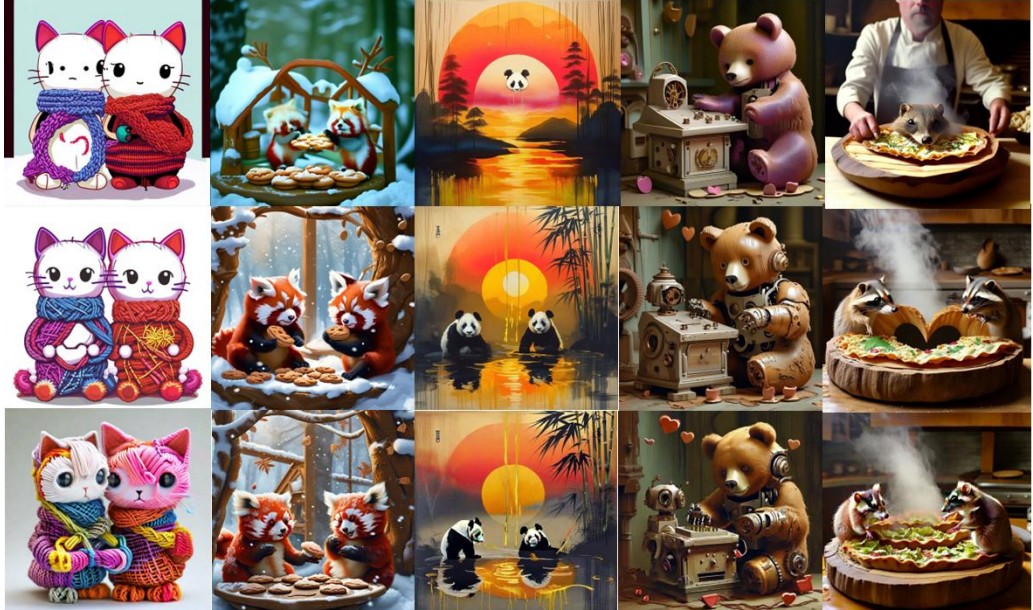

**Figure 17:** Qualitative comparison of text-to-image generation using Lumina-Next. From top to bottom: **10-step CFG**, **30-step CFG**, and **10-step RAAG**.

## E  CODE IMPLEMENTATION

In this section, we present a concise code implementation of RAAG. For all experiments in Section 4, we set $\alpha = 12$ and $w_{\max} = 18$.

```python
i = 0
while i < len(timesteps):
    t = timesteps[i]

    if self.interrupt:
        continue

    # expand the latents if we are doing classifier free guidance
    latent_model_input = torch.cat([latents] * 2) if self.
        do_classifier_free_guidance else latents
    # broadcast to batch dimension in a way that's compatible with ONNX/Core ML
    timestep = t.expand(latent_model_input.shape[0])

    latent_model_input = latent_model_input.to(prompt_embeds.dtype)
    noise_pred = self.transformer(
        hidden_states=latent_model_input,
        timestep=timestep,
        encoder_hidden_states=prompt_embeds1,
        pooled_projections=pooled_prompt_embeds,
        joint_attention_kwargs=self.joint_attention_kwargs,
        return_dict=False,
    )[0]

    # perform guidance
    if self.do_classifier_free_guidance or choice:
        noise_pred_uncond, noise_pred_text = noise_pred.chunk(2)

        if choice:
            print(f"RAAG!")

            delta = noise_pred_text - noise_pred_uncond
            uncond = noise_pred_uncond

            ratio = torch.norm(delta) / torch.norm(uncond)

            w = 1 + (w_max - 1) * torch.exp(-alpha * ratio)

            noise_pred = noise_pred_uncond + w * (noise_pred_text -
                noise_pred_uncond)
            print(f"Step {i} Guidance Scale: {w}.")

        else:
            print("org!")
            noise_pred = noise_pred_uncond + self.guidance_scale * (
                noise_pred_text - noise_pred_uncond)

    # compute the previous noisy sample x_t -> x_t-1
    latents_dtype = latents.dtype
```

```
48      latents = self.scheduler.step(noise_pred, t, latents, return_dict=False)[0]
            # if i > 0 else latents
49
50      i += 1
51
52      if latents.dtype != latents_dtype:
53          if torch.backends.mps.is_available():
54              # some platforms (eg. apple mps) misbehave due to a pytorch bug:
                    https://github.com/pytorch/pytorch/pull/99272
55              latents = latents.to(latents_dtype)
56
57      if callback_on_step_end is not None:
58          callback_kwargs = {}
59          for k in callback_on_step_end_tensor_inputs:
60              callback_kwargs[k] = locals()[k]
61          callback_outputs = callback_on_step_end(self, i, t, callback_kwargs)
62
63          latents = callback_outputs.pop("latents", latents)
64          prompt_embeds = callback_outputs.pop("prompt_embeds", prompt_embeds)
65          negative_prompt_embeds = callback_outputs.pop("negative_prompt_embeds",
                  negative_prompt_embeds)
66          negative_pooled_prompt_embeds = callback_outputs.pop(
67              "negative_pooled_prompt_embeds", negative_pooled_prompt_embeds
68          )
69
70      # call the callback, if provided
71      if i == len(timesteps) - 1 or ((i + 1) > num_warmup_steps and (i + 1) %
            self.scheduler.order == 0):
72          progress_bar.update()
73
74      if XLA_AVAILABLE:
75          xm.mark_step()
```

Listing 1: Code Implementation of RAAG.

## F  THE USE OF LARGE LANGUAGE MODELS (LLMS)

In this paper, LLM was used solely to refine text, syntax, and enhance readability. It did not contribute to anything related to the core ideas or scientific content (ideas, methods, theories, derivations, charts, results, and so on). In addition, all LLM-refined texts are manually double-checked for hallucinations and misunderstanding.