# OpenReview forum: "RAAG: RATIO AWARE ADAPTIVE GUIDANCE"
_ICLR.cc/2026/Conference — Submitted to ICLR 2026_

### Official Review · Reviewer_5JgS · 2025-11-01

**Soundness:** 3
**Presentation:** 3
**Contribution:** 3
**Rating:** 4
**Confidence:** 3

**Summary:**

This work analyses the dynamics under classifier-free guidance of flow-based models, drawing attention to the sampling instability due to a fixed guidance scale. The authors further introduce RAAG - an adaptive guidance schedule - which penalises high ratio of conditional velocity to unconditional velocity, especially during the initial phase. Much like CFG, the approach is essentially plug-and-play for flow-based models when pretrained conditional and unconditional networks are available. Experiments are provided on image benchmarks such as SD3.5 and Qwen, and video benchmark (WAN2.1) to demonstrate benefits of RAAG.

**Strengths:**

--- The motivation for this work is strong, and the overall presentation is nice.

--- Establishing the insufficiency of fixed guidance scale to data distribution (instead of a particular flow model) is appealing.

---  The authors are transparent about the limitations of the proposed approach (e.g., issues with diffusion models).

--- Sampling speedups are noteworthy.

**Weaknesses:**

--- Dynamically adjusting the ratio between the magnitude of the velocity gap and the unconditional velocity could potentially induce issues such as distributional shift (relative to data distribution) due to sampling bias. Experiments do not report widely used metrics such as FID, PSNR, Inception Score, LPIPS, precision, and recall (e.g., for CIFAR10).  This makes it objectively hard to assess the performance of the method.

--- A particularly relevant recent work [1], not discussed in the paper, has formalised why/when (adaptively) scaling the (conditional) velocity vector, including the earlier time-steps, influences the detail in image generation in flow models. CGF can be viewed as an affine combination of conditional and unconditional vector fields, and RAAG as an adaptive affine combination. However, it is unclear whether or to what extent theoretical and empirical findings with respect to RAAG are consistent with respect to, or extend beyond, the formalism and results in [1].

[1] Karczewski et al. Devil is in the Details: Density Guidance for Detail-Aware Generation with Flow Models. ICML 2025.

--- RAAG does not seem to work for diffusion models.

**Questions:**

--- Could you please quantify performance with respect to the evaluation metrics that I mentioned under the weaknesses? Ideally, you could take readily available pretrained conditional and unconditional models from a repository like NVLabs EDM, and compare RAAG with CFG across these metrics on CIFAR-10, FFHQ, ImageNet etc.

--- Could you address my other concern with respect to reference [1] under the weaknesses section?

--- Clearly, the optimal adaptive guidance scale in diffusion models does need exhibit exponential decay with respect to ratio. But how about utilising the corresponding probabilistic flow ODE? Also, perhaps a non-monotonic decay schedule (e.g., dampened cosine) might be required to extend RAAG to diffusion models.

---

### Official Review · Reviewer_zzXF · 2025-11-01

**Soundness:** 2
**Presentation:** 2
**Contribution:** 2
**Rating:** 2
**Confidence:** 3

**Summary:**

The paper focuses on classifier-free guidance, arguing that the earliest steps in flow-based generators are sensitive to the guidance scale.  It provides a theoretical analysis explaining why early over-steering harms trajectories, formalizing the effect via the gap between conditional and unconditional velocities. Building on this, the authors introduce a RATIO-aware adaptive schedule that down-weights guidance early and relaxes it later, speeding up conditional generation without degrading sample quality.

**Strengths:**

- The paper provides a theoretical explanation of the impact of early-step guidance.

- The proposed adaptive guidance schedule (Eq. 8) is closed‑form, training‑free, and easy to integrate into standard rectified flow pipelines.

**Weaknesses:**

- There are inconsistencies in core definitions or result labels (see Questions below). These inconsistencies severely affect the technical rigor of the paper and cause confusion.

- Line 90 claims 'The schedule is parameter-free at test time', while $\alpha$ and $w_{max}$ are hyperparameters for the method. It is unclear how the authors define `parameter-free' in this context.

- The paper asserts the initial RATIO is a dataset property rather than model‑dependent. However,  the measurement is only after SD3.5 VAE encoding (Appendix C.1), so it is entangled with the latent representation/model. This does not establish a model-agnostic, distribution-level inevitability.

- The theoritical results rely on restrictive assumptions (e.g. $\lambda$>0 , $\sigma$>0, ) and uses heuristic inequalities.

**Questions:**

-  Line 133 defines $v_{u}(x_{t})=E[x_0 -x_1 |x_t,\emptyset]$ and $v_{c}(x_{t},c)=E[x_1 -x_0 |x_t,c]  $. However, Eq. 3 writes $v_{c}(x_{t},c)=E[x_0 -x_1 |x_t,c]  $.  What is the correct definition?

-  Line 304 gives Proposition 3.1, while Line 435/740 mentions Theorem 3.1. What is the correct label of the results?

-  Is there any justification for the assumptions in Proposition 3.1, such as the positiveness of $\lambda$ and $\sigma$?

-  What broader cross-model/cross-dataset evidence demonstrates that the early-step RATIO spike is “inherent” to the training distribution?

- What is the relationship between the theoretically optimal solution $w=1/\rho$ and the exponential decay rule in Equation 8? Is the exponential rule near-optimal across models?

- Could the authors please explain the underperformance on classic diffusion (e.g., SD-v2)?

---

### Official Review · Reviewer_yNBK · 2025-11-02

**Soundness:** 1
**Presentation:** 1
**Contribution:** 1
**Rating:** 0
**Confidence:** 4

**Summary:**

The authors study classifier-free guidance in flow-based models. The authors focus their analysis on the relative difference of conditional and unconditional drifts and refer to it as RATIO. The authors propose an adaptive guidance schedule and demonstrate its empirical performance.

**Strengths:**

I believe that the problem that authors tackle is an important one, i.e. trying to better understand classifier-free guidance and its empirical performance.

**Weaknesses:**

1. The theoretical claims are not convincing for two reasons.

First, they are not presented well. For example, it is not even clear to me what the authors are trying to prove with their theoretical analysis. The main result is Proposition 3.1, which provides a lower bound (not really a bound, because they use the symbol $\gtrsim$, which means "greater than or approximately equal to") on the distance between two arbitrary trajectories. I don't think this is relevant to the problem of classifier-free guidance. The authors try to minimize this bound, but I don't understand why this is something that should be done or how it is supposed to solve any of the problems raised earlier in the paper.

Secondly, they do not seem to lead to practical algorithms that improve over CFG. For example, the method that directly uses the theoretical results is presented in Figure 9 and the resulting generations are clearly distorted. Therefore, the authors resort to a heuristic. The heuristic works for some models, but for some yields significantly worse results than the standard CFG. The formalism and the proof suggest that the approach should work for any flow-based model. However, tables 2 and 3 show that the proposed method, when applied to some diffusion models, is significantly worse than the standard CFG on both analyzed metrics, and the authors do not propose an explanation of this.

Therefore, I do not see merit in the proposed work. The theoretical analysis is fundamentally flawed in my opinion. The quantity analyzed (the difference between trajectories or its approximate lower bound) is not motivated and does not lead to useful algorithms.

2. The paper largely uses imprecise wording and notation. For example
    * line 79 - "Because standard pipelines apply a fixed guidance scale w throughout sampling, their effective control term of sampling errors is [...] ". What do authors mean by "control term of sampling errors? Why is it related to the ratio?
    * line 181 - "We prove this stems inherently from data distributions". What does this mean?
    * Line 315 - "which could amplify initial perturbations and increase the error lower bound" what error? Distance between trajectories is not an error
    * Line 363 - "does not guarantee reduced actual sampling error". The authors refer to a distance between trajectories as "sampling error".
    * What is $\rho$? I think the first time this symbol appears is in line 274 as $\rho_{max}$, which is not defined, and I assumed it means just some arbitrary constant defined for the purpose of the proof. Later, $\rho$ appears in the formula in Proposition 3.1 without defining what it is. Later, from context, it seems like the authors use "RATIO" and $\rho$ interchangeably. Is it supposed to depend on $x_t$? It in mathematical expressions is seems like it's constant, but I don't understand then what it refers to. There should also be two ratios. One for $x_t$ and one for $y_t$. Is it supposed to be one of those?
    * Line 364. "$Var (1/\rho) \propto 1/\rho^4$". I don't know what authors meant to say here. This is a very imprecise notation. It seems to imply that $\rho$ is a random variable (LHS) and a real number (RHS) at the same time.
    * Line 422 - "We attribute this to the inherent model corrections compensate for earlier suboptimal conditions". "conditions"? This really feels LLM generated.

**Questions:**

Please see the weaknesses I raised above. Unfortunately, I do not see how this paper can be improved. I believe the approach is fundamentally flawed, as the theoretical analysis does not seem motivated, and the resulting algorithm (after heuristic tweaks) improves performance for some models but degrades quality for others.

---

### Official Review · Reviewer_5ds5 · 2025-11-06

**Soundness:** 2
**Presentation:** 3
**Contribution:** 3
**Rating:** 6
**Confidence:** 2

**Summary:**

The paper studies the tendency of a classifier-free guidance to have strong discrepancy between the conditional and unconditional velocity. The paper proposes an adaptive weighting scheme that reduces this, and shows higher sampling quality from this.

**Strengths:**

The paper studies an interesting and novel issue in CFG. The ratio phenomena is well characterised and comprehensively presented. The results are impressive, and this work likely has lasting significance.

**Weaknesses:**

I'm not totally sold on the technical analysis of the paper. I'm not sure if the ratio is the key quantity to explain this phenomena. I wonder if it is enough to understand this solely by the ratio.

I'm not sure I get the Prop 3.1: it seems to state that different initial noises can arrive at different images. Isn't this how a denoiser is supposed to work? Why is this a bad thing?

It also seems that the theoretical analysis does not align with the empirical weight tuning, which seems a bit adhoc.

**Questions:**

See above

---

### Meta-Review · Area_Chair_51Qu · 2025-12-29

**Summary:**

This paper proposes an adaptive weighting scheduler for guidance in flow-based models based on the ratio between the conditional signal and the unconditional signal.

Reviewers commented positively on the identification of an interesting issue in CFG, the method's training‑free and easy-to-use nature, and the sampling speedups.

Reviewers commented negatively on the limited evaluation metrics in the experiments, the theoretical analysis is not convincing enough, the inconsistency between the theoretical analysis and actual implementation, and the imprecise wording and notation.

Overall, according to reviewers’ recommendations, the weaknesses outweigh the strengths.

**Reviewer Concerns:**

No rebuttal available.

**Reviewer Scores:**

No rebuttal available.

---

### Decision · Program_Chairs · 2026-01-26

Reject